# Limitations of Linear Dichroism Spectroscopy for Elucidating Structural Issues of Light-Harvesting Aggregates in Chlorosomes

**DOI:** 10.3390/molecules26040899

**Published:** 2021-02-09

**Authors:** Lisa M. Günther, Jasper Knoester, Jürgen Köhler

**Affiliations:** 1Spectroscopy of Soft Matter, University of Bayreuth, Universitätsstr. 30, 95440 Bayreuth, Germany; lisa.guenther@uni-bayreuth.de; 2University of Groningen, Zernike Institute for Advanced Materials, Nijenborgh 4, 9747 AG Groningen, The Netherlands; j.knoester@rug.nl; 3Bayreuth Institute for Macromolecular Research (BIMF), University of Bayreuth, Universitätsstr. 30, 95440 Bayreuth, Germany; 4Bavarian Polymer Institute, University of Bayreuth, Universitätsstr. 30, 95440 Bayreuth, Germany

**Keywords:** linear dichroism, molecular aggregates, light harvesting, chlorosomes, photosynthesis

## Abstract

Linear dichroism (LD) spectroscopy is a widely used technique for studying the mutual orientation of the transition-dipole moments of the electronically excited states of molecular aggregates. Often the method is applied to aggregates where detailed information about the geometrical arrangement of the monomers is lacking. However, for complex molecular assemblies where the monomers are assembled hierarchically in tiers of supramolecular structural elements, the method cannot extract well-founded information about the monomer arrangement. Here we discuss this difficulty on the example of chlorosomes, which are the light-harvesting aggregates of photosynthetic green-(non) sulfur bacteria. Chlorosomes consist of hundreds of thousands of bacteriochlorophyll molecules that self-assemble into secondary structural elements of curved lamellar or cylindrical morphology. We exploit data from polarization-resolved fluorescence-excitation spectroscopy performed on single chlorosomes for reconstructing the corresponding LD spectra. This reveals that LD spectroscopy is not suited for benchmarking structural models in particular for complex hierarchically organized molecular supramolecular assemblies.

## 1. Introduction

Green-(non-)sulfur bacteria [1,2] are known to grow photosynthetically under the lowest light conditions [3,4,5] owing to their highly efficient light-harvesting antenna, so-called chlorosomes. These are organelles enclosed by a monolipid envelope that accommodates hundreds of thousands of densely packed bacteriochlorophyll (BChl) *c*,*d*,*e* molecules, that are organized in self-assembling secondary structures featuring curved lamellar or cylindrical geometry [4,6,7]. In particular, natural chlorosomes contain mixtures of BChl homologs with modifications of the methylation status giving rise to a tremendous variety for the possible building blocks that are involved in the self-assembly process for the molecular aggregates within the chlorosomes [8,9]. Given the variations in size, shape, and the number of aggregates within a single chlorosome, it is not surprising that these molecular assemblies feature a large degree of structural heterogeneity. As a consequence of this, structure determination by high-resolution X-ray crystallography is not feasible, and chlorosomes have been investigated by various combinations of other techniques including atomic force microscopy [10,11,12], solid-state NMR [13,14], optical spectroscopy both on ensembles [11,15,16,17,18,19,20,21,22], and single-chlorosomes [12,23,24,25,26,27,28,29], cryo-electron microscopy [4,30,31], and computational approaches [6,32,33,34]. Nevertheless, details about their structural organization are still the subject of an ongoing debate.

In molecular aggregates, such as chlorosomes, the character of the electronically excited states is determined by the intermolecular interactions, which in turn are imposed by the geometrical arrangement of the molecular building blocks, i.e., the distances and mutual orientations of the monomers [35,36,37,38]. For strongly interacting monomers, the fundamental electronic excitations are delocalized over a substantial part of the aggregate and referred to as (molecular) excitons [39,40]. The formation of excitons has strong consequences for the optical spectra, causing shifts of the excitation energies and a redistribution of oscillator strengths with respect to the monomer spectra. Therefore, some information about the mutual orientation of the transition-dipole moments becomes available via spectroscopy with polarized light [41]. This allows to compare experimental spectra with predictions derived from structural models, and forms the starting point for example for linear dichroism (LD) spectroscopy, sometimes referred to as the “poor man’s crystallography” [42].

A linear dichroism (LD) spectrum corresponds to the difference of two absorption spectra that are recorded with incident light of mutually orthogonal linear polarization, i.e.,
(1)$$LD\left(\nu \right)\propto \left(A_{∥}\left(\nu \right)-A_{⊥}\left(\nu \right)\right)$$
where $A_{∥}\left(\nu \right)$ and $A_{⊥}\left(\nu \right)$ refer to the absorption spectra recorded for light that is linearly polarized parallel and perpendicular with respect to a reference direction. Obviously, the LD spectrum vanishes for a sample that consists of an ensemble of randomly oriented absorbers. Hence, measuring an LD spectrum requires a sample that features at least some degree of anisotropy for the orientation of its absorbers. This concept is illustrated in Figure 1. Consider a perfectly oriented sample with a broad, spectrally poorly resolved absorption band that is composed of two contributions from two electronic transitions with different transition energies but with transition-dipole moments of equal magnitude that are aligned parallel with respect to each other, Figure 1a top. Recording the absorption spectrum as a function of the linear polarization of the incident light for various excitation frequencies *ν* across the band yields a modulation of the absorption strength $A\left(\nu \right)\propto \cos^{2}\alpha$. Here, α refers to the angle between the projection of the transition-dipole moment of the corresponding transition on the plane perpendicular to the propagation of the incident light and the direction of the polarization of the electric field vector. Assigning $A_{∥}\left(\nu \right)$ to the maximum and $A_{⊥}\left(\nu \right)$ to the minimum of the modulation results in an LD spectrum that reproduces the absorption spectrum.

The situation changes if the transition-dipole moments of the two (or more) electronic transitions are not oriented parallel with respect to each other, Figure 1b,c top. For illustration purposes we assume that the transition-dipole moment that can be associated with the red wing of the spectrum is oriented parallel with respect to the reference direction, and vice versa, that the transition-dipole moment that can be associated with the blue wing of the spectrum is oriented perpendicular with respect to the reference direction. Then for some excitation energy $h\nu_{0}$ in-between, $h\nu_{red}$
*<*
$h\nu_{0}$
*<*
$h\nu_{blue}$, the two absorptions $A_{∥}\left(\nu_{0}\right)$ and $A_{⊥}\left(\nu_{0}\right)$ will be of equal strengths and the resulting LD spectrum will reverse its sign with a zero at $h\nu_{0}$. In other words, the relative phases of the intensity modulations of the absorption strengths will undergo a change as a function of the excitation energy. Then the added value of an LD spectrum is that it provides the evidence that more than one transition contributes to the broad unresolved absorption band. However, the LD spectrum does not allow to disentangle unambiguously the relative strengths and the exact number of the transitions that contribute to the band. Nevertheless, LD spectra can be used as a first coarse test whether a proposed structure is compatible with the experimentally observed spectra.

Recently, results from linear polarization-resolved fluorescence-excitation spectroscopy performed on individual chlorosomes from the species *Chlorobaculum (Cba.) tepidum* [43,44] have led to debates about structural models [45] designed using ensemble LD spectra reported in the literature for benchmarking [12,17,18,22,42]. Here, we reconstruct both ensemble and single-chlorosome LD spectra from the polarization-resolved fluorescence-excitation spectra reported in [43,44] for comparison with results from available ensemble LD spectroscopy. This tends us to conclude that LD spectroscopy is not suited to discriminate between different structural models of hierarchically organized supramolecular assemblies, such as chlorosomes.

This paper is organized as follows. In Section 2, motivated by the fact that tubular assemblies seem to be the dominant secondary structures in chlorosomes [43,44], we give the generic theoretical background of linear dichroism for molecular aggregates that feature a monomer arrangement with cylindrical symmetry. This is followed in Section 3 by the presentation of the reconstructed LD spectra. For the reconstruction two different methods for defining the central alignment direction are used and the results and implications of the two approaches are compared. In Section 4, the findings are discussed in the context of the existing literature, followed by a brief summary of the materials and methods in Section 5, and our conclusions in Section 6.

## 2. Theoretical Background: Linear Dichroism for Tubular Aggregates

Elucidating the design principles of molecular aggregates with tubular morphologies has attracted considerable attention in the past [46,47,48,49,50,51,52,53]. This research is motivated by the potential of such assemblies to act as highly efficient light harvesters in novel organic solar cells. For long molecular assemblies, where the monomers are arranged in cylindrical symmetry it can be shown that only three of the exciton states are optically allowed (bright states), namely one state with its transition-dipole moment parallel to the symmetry axis (${\vec{\mu }}_{1}$) and two states with mutually orthogonal transition-dipole moments perpendicular to that axis (${\vec{\mu }}_{2,a},{\vec{\mu }}_{2,b}$) [54,55]. In practice, the exciton state with the transition moment parallel to the cylinder axis typically is lowest in energy and absorbs at $h\nu_{1}$, whereas the other two states are degenerate and absorb at $h\nu_{2}$. This situation is depicted in Figure 2a,b.

As mentioned above, registering an LD spectrum requires necessarily a sample with some degree of alignment. For getting started to calculate the LD spectrum from an ensemble of tubular aggregates with an arbitrary degree of alignment, we assume that all tubes are identical and address the effects of disorder (inhomogeneity) later on. In the following, we define the z-axis as the central alignment direction, referred to as the parallel direction (∥). Then the orientation of an individual tubular molecular aggregate can be specified by the angle θ of its transition-dipole moment ${\vec{\mu }}_{1}$ with respect to the z-axis, and the angle φ between the projection of the transition-dipole moment ${\vec{\mu }}_{1}$ on the x,y plane and the x-axis, see Figure 2c. Given the degeneracy of the other two states, the relative orientation of the tubular aggregate around the ${\vec{\mu }}_{1}$ axis needs not to be specified, because any linear combination of the exciton states connected with ${\vec{\mu }}_{2,a},{\vec{\mu }}_{2,b}$ is again an eigenstate of the system. Then, the transition-dipole moments of an individual aggregate can be represented in (x,y,z) coordinates by
(2)$${\vec{\mu }}_{1}=\mu_{1}\left(\sin\theta \cos\varphi ,\;\sin\theta \sin\varphi ,\cos\theta \right)$$
(3)$${\vec{\mu }}_{2,a}=\mu_{2}\left(\cos\theta \cos\varphi ,\;\cos\theta \sin\varphi ,-\sin\theta \right)$$
(4)$${\vec{\mu }}_{2,b}=\mu_{2}\left(-\sin\varphi ,\;\cos\varphi ,0\right)$$
where $\mu_{1}$ and $\mu_{2}$ denote the magnitudes of the transition-dipole moments. The contribution of this particular aggregate to the LD spectrum can be obtained from the projections on the z- and the x-axis, respectively
(5)$$A_{∥}\left(\nu \right)=A_{1}\left(\nu \right)\cos^{2}\theta +A_{2}\left(\nu \right)\sin^{2}\theta$$
(6)$$A_{⊥}\left(\nu \right)=A_{1}\left(\nu \right)\sin^{2}\theta \cos^{2}\varphi +A_{2}\left(\nu \right)\cos^{2}\theta \cos^{2}\varphi +A_{2}\left(\nu \right)\sin^{2}\varphi$$
where $A_{1}$ ($A_{2}$) refers to the absorption spectrum associated with the oscillator strength $\mu_{1}^{2}$ ($\mu_{2}^{2}$) centred at the frequency $\nu_{1}$($\nu_{2}$), which yields for the LD spectrum
(7)$$LD\left(\nu \right)\propto A_{1}\left(\nu \right)\left(\cos^{2}\theta -\sin^{2}\theta \cos^{2}\varphi \right)+A_{2}\left(\nu \right)\left(\sin^{2}\theta -\cos^{2}\theta \cos^{2}\varphi -\sin^{2}\varphi \right)$$

The LD spectrum of an ensemble of tubular systems with an arbitrary degree of alignment that is cylindrically symmetric around the z-axis is then obtained by averaging these contributions over the orientations, Figure 2d,
(8)$$\langle LD\left(\nu \right)\rangle =C\int_{0}^{\frac{\pi }{2}}d\theta \int_{0}^{2\pi }d\varphi \sin\theta f\left(\theta \right)LD\left(\nu \right)$$

Here, f(θ) specifies the distribution of orientations of the tubes relative to the z-axis, i.e., the degree of alignment, and *C* is a normalization factor. Note that *θ* runs from 0 to π/2, because the absolute direction of the dipole does not affect the absorption strength. Inserting Equation (7) into Equation (8), and using
(9)$$C={\left(2\pi \int_{0}^{\frac{\pi }{2}}d\theta \sin\theta f\left(\theta \right)\right)}^{-1}$$
for the normalization, we obtain after integrating over the angle φ
(10)$$\langle LD\left(\nu \right)\rangle =\kappa \cdot \left(A_{1}\left(\nu \right)-A_{2}\left(\nu \right)\right)$$
with
(11)$$\kappa =\frac{\int_{0}^{\frac{\pi }{2}}d\theta \;\sin\theta \;f\left(\theta \right)\;\left[\frac{3}{2}\cos^{2}\left(\theta \right)-\frac{1}{2}\right]}{\int_{0}^{\frac{\pi }{2}}d\theta \;\sin\theta \;f\left(\theta \right)}$$

Identifying $\frac{3}{2}\cos^{2}\theta -\frac{1}{2}=P_{2}\left(\cos\theta \right)$ with the 2nd Legendre polynomial $P_{2}\left(\cos\theta \right)$, Equation (11) can be rewritten as
(12)$$\kappa =\frac{\int_{0}^{1}dq\;P_{2}\left(q\right)\;f\left(\arccos\left(q\right)\right)}{\int_{0}^{1}dq\;f\left(\arccos\left(q\right)\right)}$$

For a uniform distribution of the orientations of the tubes within a cone with top-angle $\theta_{0}$ this yields
(13)$$\kappa \left(\theta_{0}\right)=\frac{\int_{\cos\theta_{0}}^{1}dq\;\left(\frac{3}{2}q^{2}-\frac{1}{2}\right)}{\int_{\cos\theta_{0}}^{1}dq}=\frac{\frac{1}{2}\left(\cos\theta_{0}-\cos^{3}\theta_{0}\right)}{1-\cos\theta_{0}}$$

Because the spectral information is associated only with the difference of the absorption spectra $A_{1}$ and $A_{2}$, the shape of the LD spectrum is not affected by the degree of sample alignment that is represented by the factor *κ*. Hence, any method that induces an anisotropy in the alignment of the transition-dipole moments is suited for obtaining LD signals. However, the strength of the signal is scaled by the factor *κ* which reflects the quality of the alignment, Figure 2d, and which averages out to zero for isotropic samples, Figure 2e. It is straightforward to extend these considerations to an ensemble of tubular assemblies with different transition energies (inhomogeneous broadening) as long as the degree of alignment is not correlated with the energetic disorder. Then the experimentally observed ensemble LD spectrum corresponds to the difference of the individual absorption spectra $A_{1}\left(\nu \right)$ and $A_{2}\left(\nu \right)$ averaged over the relevant disorder
(14)$$\bar{\langle LD\left(\nu \right)\rangle }=\kappa \cdot \left(\bar{A_{1}\left(\nu \right)}-\bar{A_{2}\left(\nu \right)}\right)$$

## 3. Results: Linear Dichroism of Chlorosomes

In a conventional LD experiment the chlorosomes feature a macroscopic preferential alignment that has been induced by an external procedure, for example using squeezing techniques, or involving electric or magnetic fields [41,42]. In order to define an alignment direction for reconstructing LD spectra from the polarization-resolved fluorescence-excitation spectra, we determined the modulation of the fluorescence as a function of the polarization at the spectral peak position, $h\nu_{peak}$, of the polarization-averaged spectrum of the particular single chlorosome. This is illustrated in Figure 3 for two individual chlorosomes from the *bchR* mutant of the species *Chlorobaculum (Cba.) tepidum*. Figure 3a,b shows two examples of 200 consecutively recorded polarization-resolved fluorescence-excitation spectra in a two-dimensional representation. The horizontal axes correspond to the excitation energy, the vertical axes correspond to the polarization of the incident radiation, and the resulting fluorescence intensity is given by the colour code. Between the recordings of two subsequent spectra the linear polarization of the excitation light was rotated by 3°. The vertical green line refers to the position, $h\nu_{peak}$, of the respective maximum of the polarization averaged intensity, and the intensity modulation at this position as a function of the polarization is shown next to the pattern. Then, the angle (modulo 180°) for which this modulation featured a maximum was defined as the central alignment direction (“parallel”, ∥). For the examples shown in Figure 3a,b this yields $\alpha_{∥}$ = 69.7° (modulo 180°), and 11.2° (modulo 180°), respectively. The corresponding $\alpha_{⊥}$ is then obtained by adding 90° to these angles. For obtaining the LD spectrum, ($A_{∥}\left(\nu \right)-A_{⊥}\left(\nu \right)$), the contributions from all individual excitation spectra registered for polarizations $\alpha_{∥}$ + *n*⋅180° and $\alpha_{⊥}$+ *n*⋅180° (*n* integer) were averaged and subtracted from each other, Figure 3c,d. This protocol relies on the assumption that the incorporation of the secondary structural elements into the individual chlorosomes follows roughly the same overall organizational scheme. This is supported by data from cryo-electron microscopy [4,7], where it appears that the tubular structures are typically aligned along the long direction of the chlorosomes, which is also in line with the observation of the strong modulations of the fluorescence intensity as a function of the polarization [12,23,28,29,43,44]. As will be shown below the presence of such an organization is consistent with our analysis for the mutants and to a lesser extent for the WT.

In the following, this reconstruction protocol will be referred to as ‘method 1’. It allowed us to reconstruct LD spectra from 72 individual chlorosomes from the WT, 66 individual chlorosomes from the *bchR* mutant, and 29 individual chlorosomes from the *bchQR* mutant. The resulting reconstructed LD spectra will be discussed after having introduced an alternative protocol, ‘method 2’, for defining the central alignment direction.

In [43], a fit routine was used that decomposed the patterns of the polarization-resolved fluorescence-excitation spectra as shown in Figure 4 into four spectral bands of Gaussian shape, labelled $A_{1},\ldots ,A_{4}$ in the order of increasing energy. For the chlorosomes from the mutants, the four bands could be grouped into a low-energy pair ($A_{1}$ and $A_{2}$) and a high-energy pair ($A_{3}$ and $A_{4}$) with about mutually orthogonal transition-dipole moments within each pair, and about pairwise parallel transition-dipole moments for the high-energy ($A_{1}$ and $A_{3}$) and low-energy ($A_{2}$ and $A_{4}$) components of each pair, respectively.

Since it is known from AFM experiments that the chlorosomes are oriented with their long axis parallel to the substrate [10,11,12] we deduced a cylindrical symmetry for the arrangement of the monomers from the polarization behaviour of the bands $A_{1},\ldots ,A_{4}$. This conclusion was in agreement with results obtained from cryo-electron microscopy [4,7,13]. For the WT chlorosomes we found two sub-populations, WT-group1 (74 %) and WT-group2 (26%). The spectral bands from WT-group1 could be grouped as well into a low- and a high-energy pair of transitions with polarization properties agreeing with an underlying cylindrical symmetry of the monomer arrangement. However, the distributions for the phase differences between the bands $A_{1},\ldots ,A_{4}$ were clearly broader than the corresponding distributions from the mutants. For the second population of WT chlorosomes, WT-group2, the bands $A_{1},\ldots ,A_{4}$ featured about equidistant spectral separation and a classification into a low- or high-energy pair was not meaningful. For these chlorosomes, the transition-dipole moments that were associated with the bands $A_{1}$, $A_{2}$, and $A_{4}$ were oriented about parallel with respect to each other, and about perpendicular with respect to the transition-dipole moment that was associated with the band $A_{3}$. For WT-group-2 chlorosomes, the structure that underlies the polarization properties still is unclear.

For those chlorosomes whose spectral signatures are consistent with a monomer arrangement in cylindrical symmetry—i.e., for *bchR*, *bchQR*, and WT-group1—the above considerations leading to Equation (14) can be generalized to the situation encountered here, i.e., for two parallel transitions and four pair-wise degenerate perpendicular ones. Accordingly, the transition-dipole moments of the low-energy transitions, $A_{1}$ and $A_{3}$, are oriented parallel with respect to the symmetry axis of the tubular structures providing an unambiguous criterion for defining the central alignment direction (“parallel”, ∥) of the tubular secondary structural elements, referred to as ‘method 2’ in the following.

In Figure 5 the reconstructed LD spectra are shown for ‘method 1’ by the blue lines, and for ‘method 2’ by the grey lines. For those individual chlorosomes where both methods could be applied, the corresponding LD spectra are overlaid. The LD spectra at the bottom of each column correspond to the sum of the respective single-chlorosome spectra and represent an ‘ensemble’ spectrum.

In general, the LD spectra reveal large differences in the widths of the spectral features for the three types of chlorosomes. While the spectra from the two mutants cover a spectral range of about 400 cm^−1^ the spectra from the WT cover about twice this range. This observation is in agreement with previous work, where we observed similar differences for the widths of the absorption bands from the three species [44], and which simply reflects the reduction of the structural heterogeneity by decreasing the BChl homolog sidechain heterogeneity employing mutagenesis [56]. Pairwise comparison of the LD spectra reconstructed using the two different methods reveals that these can be categorized into those that show good agreement (A), good agreement but a sign flip (B), reasonable agreement (C), and those that do not agree (D) with each other, as exemplified in Figure 5 from top to bottom, and summarized in Table 1.

As can be shown, see supporting information, the two reconstruction methods give the same result (except for a possible sign flip), if the transition-dipole moments associated with the low-energy pair $A_{1}$, and $A_{2}$ (high-energy pair, $A_{3}$, and $A_{4}$) are oriented exactly perpendicular with respect to each other, and if the transition-dipole moments of transitions associated with $A_{1}$ and $A_{3}$ (and concomitantly $A_{2}$ and $A_{4}$) are oriented parallel with respect to each other. Hence, merging the categories A, B, and C that gave reasonable or even good agreement for the results from both methods, it can be deduced that for the vast majority of the chlorosomes studied both the molecular packing within the secondary structural elements as well as the mutual alignment of the secondary structures with respect to each other is very regular. On the other hand, the results from the two reconstruction methods will disagree for significant deviations of the involved transition-dipole moments from the above mentioned ‘ideal’ geometry. We may speculate that this in turn originates from some misalignment of the tubular aggregates inside the chlorosomes, or alternatively from structural or energetic disorder within the tubular aggregates that breaks the strict selection rules for optical transitions and allows for polarization angles that deviate from 0° and 90° [57].

Moreover, the LD spectra from individual chlorosomes feature clear variations in shape irrespective of the reconstruction method. This is summarized in Table 2, where we distinguish LD spectra with no node as type 1 (+ or −), with one node as type 2 (+ − or − +), and with two nodes as type 3 (+ − + or − + −). For those chlorosomes where both LD reconstruction methods could be applied the occurrence of the different types of LD spectra for the same type of chlorosome can be compared, see Table 2. Given the limited statistics in particular for the *bchQR* mutant, this reveals that the abundancies of the different types of LD spectra for both reconstruction protocols are similar within statistical error. From four pairwise, perfectly perpendicular polarized bands one would expect to detect an LD spectrum with three nodes. However, even for this ideal situation, it will depend on the spectral separation, the relative intensities, and the widths of these bands whether these nodes can be resolved experimentally. In general, smaller spectral separation and/or larger linewidths will blur details in the spectra and suppress the observation of nodes (see also Appendix A). This is reflected in the relative abundancies of the different types of LD spectra across the different types of chlorosomes. The occurrence of LD spectra with one or two nodes increases slightly in the order WT < *bchR* < *bchQR*, see Table 2, which is the same order in which the widths of the absorption bands decrease [44], reflecting the (spectral) heterogeneity of the chlorosomes. In any case, most of the reconstructed LD spectra correspond to those without a node. Therefore, it is not surprising that the more complex LD spectra are fully averaged out in the ensemble LD spectra.

## 4. Discussion

In an experiment on a large ensemble of chlorosomes the macroscopic preferential alignment is induced externally. This, however, refers to the alignment of the chlorosomes, whereas the optical response to polarized light originates from the secondary structures within the chlorosomes. For the reconstruction of the LD spectra, we used a definition for the central alignment direction that was either based on the modulation of the fluorescence response at the maximum of the absorption spectrum of the respective chlorosome, or by taking advantage of the underlying symmetry properties of the spectral signatures (for those chlorosomes whose spectral signatures complied with cylindrical symmetry of the monomer arrangement). These definitions, however, are both related to the alignment of the secondary structures within the chlorosomes irrespective of their orientation relative to the chlorosome. Hence, single-chlorosome and ensemble LD spectroscopy are supposed to give equivalent results (apart from a possible sign flip) only if the secondary structural elements are incorporated into the chlorosomes according to a unique scheme, for instance if the long axis of the tubular aggregates is always parallel (or always perpendicular) to the long axis of the chlorosome.

To the best of our knowledge, LD spectra from the *bchR* and *bchQR* mutants from the green-sulfur bacterium *Cba. tepidum* have not been published before, and the only LD spectra from this species that have been reported so far are those from the WT documented in the supporting information of [12] and in the PhD thesis by de Ruijter [58]. In Figure 6a these spectra are compared with the reconstructed ensemble LD spectrum (method 1) from the current work. The reconstructed LD spectrum and the one from [12] feature about the same width but differ with respect to the spectral position of the maxima and the slope of the low energy wing, whereas the LD spectrum from [58] features about the same spectral position for the maximum but is clearly narrower than the reconstructed one. The spectrum from [12] has been obtained at room temperature on chlorosomes that were immobilized in a highly polar polyacrylamide gel, whereas the spectra shown in [58] have been acquired at 1.5 K from chlorosomes that were not embedded in a matrix. In addition, these two studies used chlorosomes that stemmed from different sources, and it is generally argued that the observed structural variations of the chlorosomes might depend on growth and sample preparation conditions [23,25,27,28,59,60]. Since the two conventional LD spectra show already large variations with respect to each other, it is difficult to draw conclusions about the origin of the discrepancies between the reconstructed LD spectrum and the ensemble LD spectra. Therefore, we extended the comparison also to LD spectra from other species of green-(non-)sulfur bacteria, namely from the green-sulfur bacterium *Prosthecochloris aestuarii* [58] recorded at 1.5 K, and LD spectra from the green non-sulfur bacterium *Chloroflexus aurantiacus* recorded both at 77 K and room temperature [17,18,19,20,21,22,61], Figure 6b,c. All these spectra resemble in shape and width the one taken by de Ruijter on *Cba. tepidum*, Figure 6a [58]. Therefore, we ascribe the different appearance of the spectrum from [12] to matrix effects and exclude this spectrum from the further discussion.

Visual inspection of the reconstructed ensemble LD spectra from the mutants shown at the bottom of Figure 5c,d reveals that these spectra, apart from being clearly narrower, resemble in shape the other conventional ensemble LD spectra. As mentioned above the reduced width of these spectra reflects the diminished sample heterogeneity of the mutants. For better comparison of the spectral profiles, we multiplied the widths of the mutant LD spectra with a factor of 2.5 and overlaid them with the available conventional LD spectra from the literature, Figure 6d. The value of 2.5 was chosen because it gave the best match for the comparison. We note that this value is also reasonably close to the factors of about 1.8 that were found for the reduction of the inhomogeneous linewidths both for the ensemble absorption spectra and the single chlorosome excitation spectra from *Cba. tepidum* upon mutagenesis [43,44]. These comparisons yield nearly a perfect match, providing strong evidence that the discrepancy between the reconstructed WT ensemble spectrum and the ensemble LD spectrum from [58] observed in Figure 6a is ‘real’, and cannot be ascribed to the limited statistics.

As we have pointed out above, single-chlorosome and ensemble LD spectroscopy are supposed to give equivalent results only for the case that the secondary structural elements are incorporated into the chlorosomes according to a unique scheme. This tends us to conclude that this prerequisite is well fulfilled for the mutants and only in part for the WT chlorosomes. This is in agreement with cryo-electron microscopy data obtained on chlorosomes from *Cba. tepidum*, where a “more regular cylindrical organization” was found for the secondary structures of a mutant, whereas for the WT the tubular structures were “embedded in a matrix of structurally less defined assemblies” [4,7,13]. Given the high degree of internal organization for the chlorosomes from the mutants and in part for the WT our findings support the speculation made earlier that for these chlorosomes the secondary structures extend through the full length of the chlorosomes [7].

For WT-group2 the polarization properties of the bands $A_{1}$ and $A_{2}$ do clearly not comply with an underlying cylindrical symmetry for the secondary structural elements, whereas the mutual orthogonal polarization of the bands $A_{3}$ and $A_{4}$ does at least not exclude this morphology. As has been shown in [12], mutually orthogonal transition-dipole moments do not necessarily require closed tubular molecular assemblies. An anisotropy that allows to define a preferential axis, as for example for a quarter of a cylinder or a rolled lamella, is already sufficient to obtain to a good approximation such spectral signatures. We may speculate that the less defined assemblies observed in cryo-electron microscopy might arise from WT-group2 chlorosomes, and that such chlorosomes contain a mixture of structural elements, some resembling lamella that are rolled up to different extents, others that are more similar to undulated sheets. Furthermore, it might be possible that the differences between the two groups of WT chlorosomes are associated with the growing stage of the bacteria, and that there is a gradual transition between WT-group2 and WT-group1 chlorosomes.

## 5. Materials and Methods

The experimental data used in this paper for reconstructing the LD spectra have been obtained from samples that were prepared by Don Bryant and coworkers (The Pennsylvania State University, State College, PA, USA) [2,9,59,60,61], and have been published in [43,44]. All experiments have been conducted at 1.2 K. Detailed descriptions of the experimental protocols and procedures are given in the references.

## 6. Conclusions

For the comparison of the polarization-resolved spectroscopy on individual chlorosomes with ensemble LD spectroscopy we disposed over detailed information about the single-chlorosome absorption bands beforehand. Taking advantage of this information brought us to the conclusion that the molecular packing within the secondary structures as well as the overall alignment of the secondary structures within the chlorosomes is reasonably uniform for the mutants, and less uniform for the WT.

In general, the reconstruction of the LD spectra leads to an inherent loss of information with respect to the polarization-resolved fluorescence-excitation spectra, and it turns out that for complex molecular aggregates the information that can be extracted from LD spectroscopy alone is rather limited. Commonly, LD spectroscopy cannot disentangle the number and relative strengths of multiple transitions that contribute to an absorption band. In particular for the chlorosomes, it has to be taken into account that in an experiment on a macroscopic sample the central alignment direction refers to the alignment of the chlorosomes, rather than to the alignment of the secondary structures within the chlorosomes. However, the optical response to polarized light stems solely from the secondary structural elements within the chlorosomes, whose precise alignment with respect to the long axis of the chlorosomes is not necessarily uniform. Hence, based on LD spectroscopy on ensembles of chlorosomes alone, it is extremely difficult to obtain reliable information about the mutual orientation of transition-dipole moments of the BChl monomers within the chlorosomes. Altogether this explains the discrepancies between the numerical values that were proposed for the angle between the transition-dipole moments of the BChl monomers and the symmetry axis of the secondary structures, ranging from 15–40° [17,18,19,20,21,22,58,61], and the value of about 55° that was found in [43,44], and which is in very close agreement with what has been found for a biomimetic model system [49]. For the individual chlorosomes, the reconstruction of the LD spectra uncovers clear differences in the spectral profiles with respect to widths and spectral shape, which were all averaged out in the corresponding ensemble LD spectra. It appears that LD spectroscopy alone is not suited to be used for benchmarking structural models of more intricate molecular assemblies.

## Figures and Tables

**Figure 1 molecules-26-00899-f001:**
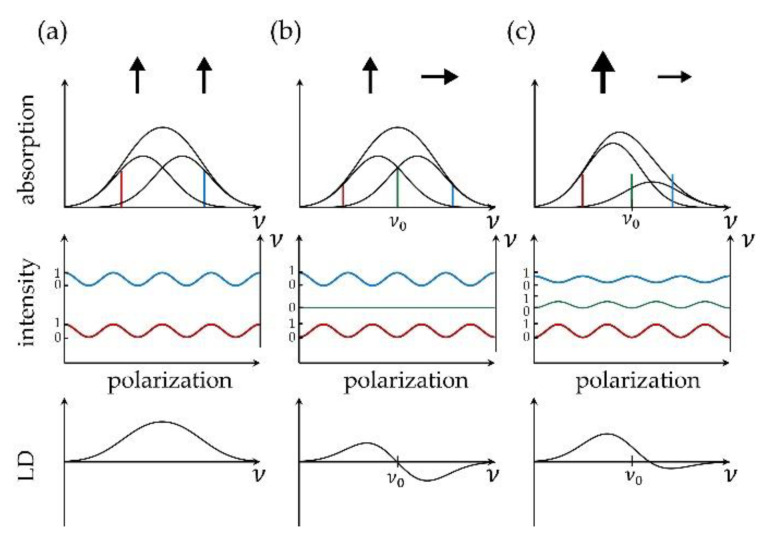
Each column (**a**–**c**) shows from top to bottom a schematic sketch of an absorption spectrum, the normalized intensity modulation as a function of the polarization for distinct spectral components, and the resulting LD spectrum. For illustration, a few transition frequencies (blue, green, red) have been indicated. The illustrations refer to perfectly oriented samples that feature (**a**) two transitions with different energies and parallel transition-dipole moments; (**b**) two transitions with different energies and mutually orthogonal transition-dipole moments of equal magnitude; (**c**) two transitions with different energies and mutually orthogonal transition-dipole moments of non-equal magnitude.

**Figure 2 molecules-26-00899-f002:**
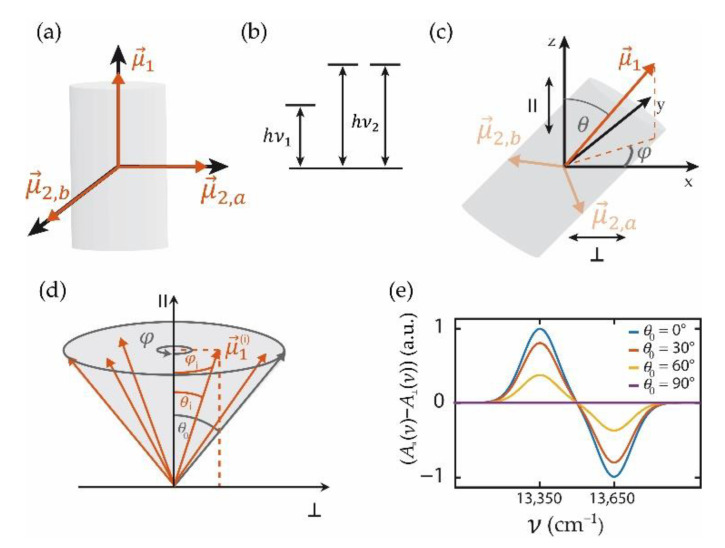
(**a**) Mutual orientation of the transition-dipole moments with non-vanishing oscillator strengths for a molecular aggregate with a monomer arrangement in cylindrical symmetry. (**b**) Excitation energies of the exciton states associated with the dipole-allowed transitions. (**c**) Relative orientation of the transition-dipole moment specified by the angles θ and φ with respect to a cartesian reference frame that defines the central alignment direction “parallel” (∥) along the z-axis. (**d**) Schematic sketch of the relative orientations of an ensemble of tubular aggregates, if the angle θ between the transition-dipole moment ${\vec{\mu }}_{1}$ and the z-axis is allowed to vary within a cone of opening angle $\theta_{0}$. (**e**) Calculated ensemble LD spectrum 〈LD(ν)〉 for cylindrical aggregates with a uniform distribution of the orientation within a cone with opening angle $\theta_{0}$ using Equation (13). For the calculation, the magnitudes of the transition-dipole moments were put on the same level, and the transitions were modelled as Gaussians with a width of 100 cm^−1^ positioned at $h\nu_{1}$ = 13,350 cm^−1^ for the transition associated with ${\vec{\mu }}_{1}$ and $h\nu_{2}$ = 13,650 cm^−1^ for the transitions associated with ${\vec{\mu }}_{2a,b}$.

**Figure 3 molecules-26-00899-f003:**
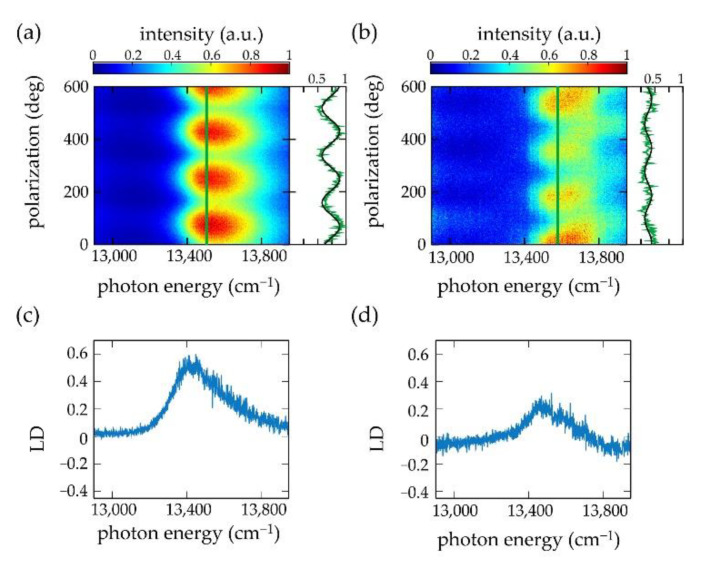
(**a**,**b**) Polarization-resolved fluorescence-excitation spectrum of two separate individual chlorosomes from the *bchR* mutant of *Cba. tepidum* at low temperature (1.5 K). On the right-hand side of the patterns the measured fluorescence intensity (green line) is compared to a cos^2^ function (black line) as a function of the polarization of the excitation light at the excitation energy of 13,490 cm^−1^ (**a**), and 13,576 cm^−1^ (**b**), respectively, corresponding to the peak energy, $h\nu_{peak}$, of the polarization-averaged spectra indicated in the pattern by the vertical green lines. Note that in (**b**) the reduction of the intensity between about 100° and 400° does not affect the spectral shape of the excitation spectrum. Also, for this chlorosome, the spectral profile is reproduced every 180°. (**c**,**d**) Reconstructed LD spectra for the two individual chlorosomes.

**Figure 4 molecules-26-00899-f004:**
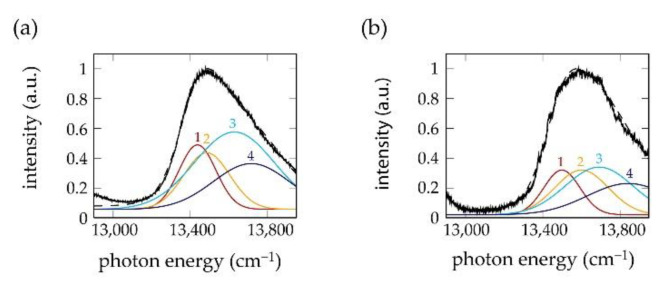
(**a**,**b**) Decomposition of the polarization-resolved fluorescence-excitation spectra from Figure 3 into four Gaussians 1 to 4 shown in yellow, red, cyan and dark blue, together with the spectrum averaged over all polarizations (black). The data have been taken from [43].

**Figure 5 molecules-26-00899-f005:**
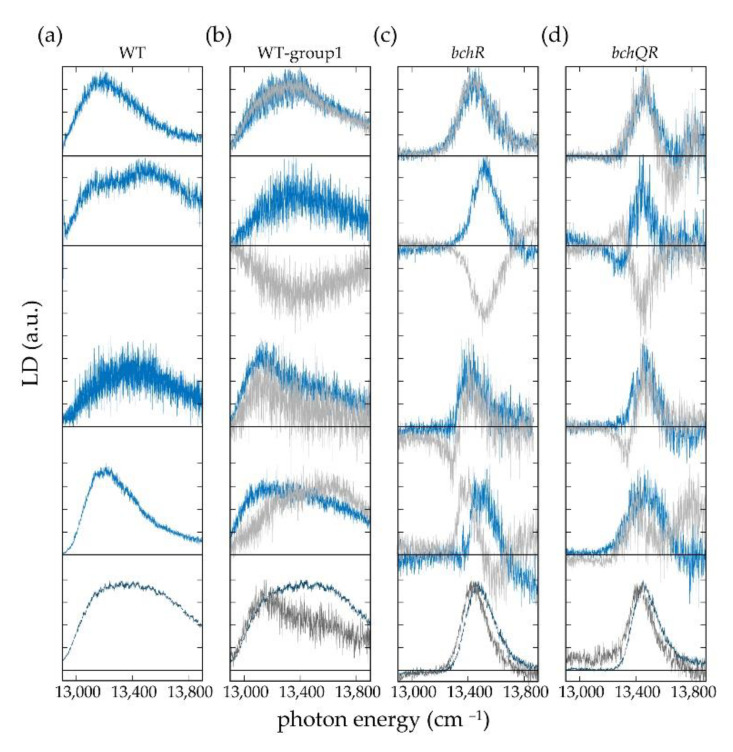
Reconstructed LD spectra using the central alignment direction obtained from the modulation of the fluorescence intensity at the maximum of the individual spectrum (light blue), or from the modulation of the Gaussian A_1_ (light grey). The panel shows from left to right examples of LD spectra of individual chlorosomes from (**a**) WT, (**b**) WT-group1, (**c**) *bchR*, and (**d**) *bchQR*. The panel shows from top to bottom examples of LD spectra from individual chlorosomes that feature good agreement (A), good agreement but a sign flip (B), reasonable agreement (C), and no agreement for the two reconstruction methods (D). A–D refer to the classification used in Table 1. The spectra shown at the bottom in dark blue (dark grey) represent the corresponding ensemble spectra obtained by summing all individual LD spectra measured for chlorosomes of types (**a**–**d**) obtained according to method 1 (method 2). For each spectrum, the zero level is given by the horizontal line, and all spectra have been normalized for better comparison. Note in passing: For the WT chlorosomes, the comparison of the two reconstruction methods is not meaningful, because the spectral signatures of the WT-group2 chlorosomes, which account for about 26% of the total, do not fulfil the prerequisite of a tubular symmetry for the underlying molecular packing.

**Figure 6 molecules-26-00899-f006:**
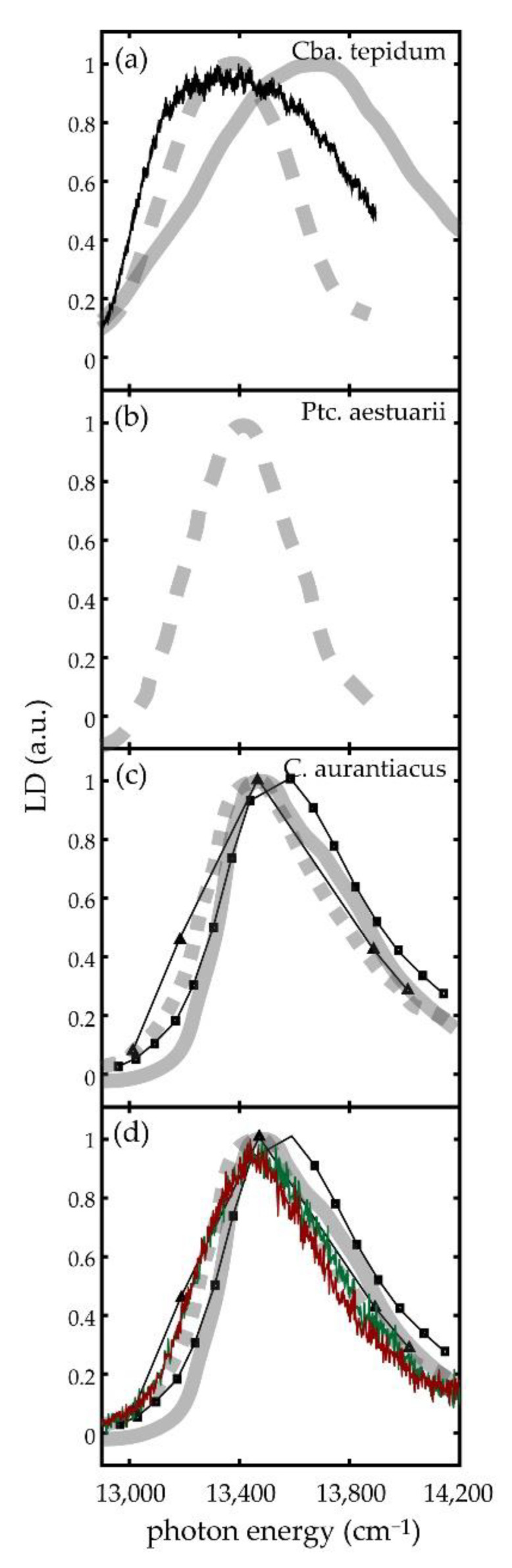
(**a**) Comparison of ensemble LD spectra from chlorosomes from the WT of *Cba. tepidum.* The noisy line (black) corresponds to the sum LD spectrum from 72 individual chlorosomes, the dashed line to the spectrum from [58], and the full line to the spectrum from ref [12]. (**b**) Ensemble LD spectrum from the WT of *Ptc. aestuarii* from [58]. (**c**) Ensemble LD spectra from the WT of *C. aurantiacus*. The spectra have been taken from [17,18,19,20,21,22,61] For better comparison all spectra have been normalized. (**d**) Same spectra as in (**c**) overlaid with the reconstructed ensemble spectra (method 1) for the *bchR* (green) and *bchQR* (red) mutants from *Cba. tepidum*. For facilitating this comparison, the widths of the reconstructed spectra have been multiplied by factor of 2.5, which gave the best agreement with the ensemble LD spectra from the WT of *C. aurantiacus*.

**Table 1 molecules-26-00899-t001:** Summary of the pairwise comparison of the single-chlorosome LD spectra that were reconstructed according to the two different protocols. The absolute number of complexes is given in parentheses.

	A	B	C	D
	Good Agreement	Good Agreement (But Sign Flip)	Reasonable Agreement	No Agreement
WT-group1	42% (22)	32% (17)	11% (6)	15% (8)
		85% (45)		
*bchR*	62% (41)	27% (18)	6% (4)	5% (3)
		95% (63)		
*bchQR*	35% (10)	41% (12)	10% (3)	14% (4)
		86% (25)		

**Table 2 molecules-26-00899-t002:** Summary of the different types of single-chlorosome LD spectra. The colour code refers to the colour code used in Figure 5 for distinguishing the two reconstruction protocols, method 1 (blue) and method 2 (grey).

	Type 1: No Node		Type 2: One Node		Type 3: Two Nodes		Relative Abundance T1:T2:T3
LD Type	+	−	+ −	− +	+ − +	− + −	
WT							
	68		4		0		94%: 6%: 0%
method 1	67	1	1	3	0	0	
WT-group1 Sign flips 32% (17)							
	49		4		0		92%:8%:0%
method 1	48	1	1	3	0	0	
method 2	27	15	7	3	1	0	
	42		10		1		79%: 19%:2%
WT-group2							
	19		0		0		100%:0%:0%
method 1	19	0	0	0	0	0	
*bchR* mutant sign flips 27% (18)							
	52		14		0		79%:21%:0%
method 1	52	0	4	10	0	0	
method 2	40	11	12	3	0	0	
	51		15		0		77%:23%:0%
*bchQR* mutant sign flips 41% (12)							
	21		6		2		72%:21%:7%
method 1	21	0	2	4	2	0	
method 2	10	7	5	4	2	1	
	17		9		3		59%:31%:10%

## Data Availability

Data is contained within the article or supplementary material. The data presented in this study are available in [Günther, L.M.; et al. *J. Phys. Chem. B*
**2016**, *120*, 5367–5376, and Günther, L.M.; et al. *J. Phys. Chem. B*
**2018**, *122*, 6712–6723.].

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
