# Peer review of "Limitations of Linear Dichroism Spectroscopy for Elucidating Structural Issues of Light-Harvesting Aggregates in Chlorosomes"

_molecules, 2021, doi:10.3390/molecules26040899_

Round 1

Reviewer 1 Report

Günther et al. present a comprehensive reexamination of linear dichroism (LD) experiments performed mainly by their group on light harvesting systems of photosynthetic bacteria, and compare analysis of single-particle spectroscopy with ensemble LD measurements. They find that depending on the analysis method chosen the single-particle spectra of chlorosomes can differ, but overall there is reasonable agreement with the ensemble spectra for those chlorosomes which show presumably intact hierarchy of the inner structure. The manuscript also presents a very useful overview of the theory of LD for cylindrical structures, and reiterates importantly that the amount of disorder does not affect the shape of the LD spectra. The text is well written and logically organized, and would make an important contribution to the ongoing debate of the structure and photophysics of this part of photosynthetic apparatus. I would recommend publication after the authors answer the comments below.

  1. I would welcome more discussion on the physical meaning of some of the findings and their implications for the chlorosome structure. Specifically, what structure would give rise to the spectral features observed for the WT-group2? Similarly, what are the chlorosomes of the group D for which the two methods give different results?
  2. Related to the previous point, might any of these effects come from structural damage of the samples themselves? How close to in-vivo state are the chlorosomes measured?
  3. Also related to the first point, what is the physical meaning and structural implication of the varying numbers of nodes observed in some of the spectra?
  4. At page 12 the authors mention that to compare single-particle LD spectra of the mutants with ensemble ones, they multiplied them (on energy scale I assume) by a factor of 2.5. How has this number been chosen? This would also imply a significant extent of inhomogeneous broadening, has this been observed in other type of spectra?

Author Response

see file attached

Reviewer 2 Report

I carefully read the paper you submitted and checked it out.

Here are my comments:

The authors showed theory of LD spectrum for chlorosomes, experimental results, and its interpretation, and discussed limitation of LD spectrum for their study.

Development of the theory was reasonable, the interpretation of the experiment was detailed, and the conclusion was acceptable. However, there was one point I would like to confirm.

Fig.3 shows two polarization-resolved fluorescence excitation spectra of sample chlorosomes. I think, in general, a value in polarization spectrum returns to original at 360 degrees. In Fig.3(a) it is reproduced, but in Fig.3(b) a value at 0 degree and that at 360 degree are difference. Rather, it seems to return at 540 degrees. My questions are why it doesn’t return at 360 degrees, why it seems to return at 540 degrees, and how about the effect of it on the reconstructions of LD spectra.

(If it has already discussed author’s previous papers or I misunderstand author's measurement principle, please tell me.)

Author Response

see file attached

Reviewer 3 Report

The manuscript by Günther et al. investigated the potential limitation of linear dichroism spectroscopy on single molecule and ensemble spectroscopy of chorosomes from wild type and mutant strains. The manuscript could have high relevance in reconstructing LD spectra from polarized fluorescence excitation spectra in chlorosomes, and in this way further information about the structure of chlorosomes could be deducted. However, in its current form the manuscript needs revisions and some methodological aspects also need clarifications.

The theoretical introduction and backgound on linear dichroism are appropriate, but the aim of the study is not described in the Introduction. Chlorosomes and its structural/functional properties should be introduced as well.

Section from line 149 regarding the chlorosomes: This section should be part of Introduction (or Discussion), not the Results.

The aims are described only in the lines 171-174, but this should be rather added to the end of Introduction.

Reconstruction protocols method 1 and method 2 should be described in details in Materials and Methods.

Discussion: the authors describe and analyze in the discussion that single chlorosome and ensemble LD spectra are in good agreement in the mutants but only in partial agreement for the WT chlorosomes. Furthermore, it is also described that the orientation methods and sample preparation and sample sources also strongly influence structural properties of chlorosomes. However, it is somewhat unclear why this is assigned to the limitations of LD spectroscopy, as the title and abstract suggests. From the comprehensive analysis it rather appears that combination of polarized fluorescence spectroscopy, linear dichroism and cryoelectron microscopy are required to elucidate the variations of the structure of the chlorosomes, and LD does not seem to be an inferior method in this aspect. Moreover, possibly several orientation methods should be compared, i.e. the authors do not mention the orientation with strong magnetic field and how this might influence the polarized spectroscopy analysis as compared with gel squeezing methods.

Materials and Methods: this section is rather incomplete, although the experimental details are referred from earlier publications, the reconstruction methods should be described here in high details, as this is crucial for the presented analyses.

Author Response

see file attached

Round 2

Reviewer 1 Report

The authors succeeded in my opinion to answer the criticism raised by all reviewers and I would recommend acceptance in the current form

Reviewer 3 Report

The authors have revised the manuscript sufficiently.